# linus: Conveniently explore, share, and present large-scale biological trajectory data in a web browser

**Johannes Waschke**[1,2], **Mario Hlawitschka**[2], **Kerim Anlas**[3], **Vikas Trivedi**[3,4], **Ingo Roeder**[5,6], **Jan Huisken**[7], **Nico Scherf**[1,6,8]*

1 Max Planck Institute for Human Cognitive and Brain Sciences, Leipzig, Germany, 2 Faculty of Computer Science and Media, Leipzig University of Applied Sciences, Leipzig, Germany, 3 EMBL Barcelona, Barcelona, Spain, 4 EMBL Heidelberg, Developmental Biology Unit, Heidelberg, Germany, 5 National Center of Tumor Diseases (NCT), Partner Site Dresden, Dresden, Germany, 6 Institute for Medical Informatics and Biometry, Carl Gustav Carus Faculty of Medicine, TU Dresden, Dresden, Germany, 7 Morgridge Institute for Research, Madison, Wisconsin, United States of America, 8 Center for Scalable Data Analytics and Artificial Intelligence ScaDS.AI, Dresden/Leipzig, Leipzig, Germany

* nscherf@cbs.mpg.de

**Data Availability Statement:** Example visualisations are available by scanning the QR codes in the figures of this manuscript directly or by visiting https://imb-dev.gitlab.io/linus-manuscript/. The software, including source code

## Abstract

In biology, we are often confronted with information-rich, large-scale trajectory data, but exploring and communicating patterns in such data can be a cumbersome task. Ideally, the data should be wrapped with an interactive visualisation in one concise packet that makes it straightforward to create and test hypotheses collaboratively. To address these challenges, we have developed a tool, *linus*, which makes the process of exploring and sharing 3D trajectories as easy as browsing a website. We provide a python script that reads trajectory data, enriches them with additional features such as edge bundling or custom axes, and generates an interactive web-based visualisation that can be shared online. *linus* facilitates the collaborative discovery of patterns in complex trajectory data.

## Author summary

Many of the processes that we study in biology are dynamic or interconnected. We can represent most of them as trajectories, being it connections between neurons in a brain or species in an ecosystem or motion traces of animals, cells or molecules. Modern experiments allow researchers to generate such trajectory data at unprecedented scales: think the parallel tracking of thousands of cells in a developing embryo over hours or days. However, visualising large-scale trajectory data is a challenge: the typical static visualisations result in excessive overplotting and often resemble the infamous hairballs. Simplification and interactivity are crucial strategies to deal with this problem. We present the lightweight tool *linus* that enables researchers to explore and share their trajectory data in an engaging way in web browsers from almost any device.

This is a *PLOS Computational Biology* Software paper.

and documentation, is freely available at our repository at https://gitlab.com/imb-dev/linus.

**Funding:** J.W. received funding from the International Max Planck Research School on Neuroscience of Communication: Function, Structure, and Plasticity (Leipzig, Germany; https://imprs-neurocom.mpg.de). K.A. and V.T. acknowledge funding from European Molecular Biology Laboratory (EMBL) Barcelona and Mesoscopic Imaging Facility, EMBL Barcelona for help with imaging. The funders had no role in study design, data collection and analysis, decision to publish, or preparation of the manuscript.

**Competing interests:** The authors have declared that no competing interests exist.

## Introduction

In biology, we often face large-scale trajectory data from dense spatial pathways, such as the brain connectivity obtained from diffusion MRI imaging [1], or tracking data such as cell trajectories [2] or animal trails [3]. Although this type of data is becoming increasingly prominent in biomedical research [4–6], exploring, sharing, and communicating patterns in such data are often cumbersome tasks requiring a set of different software that are often complex to install, learn and use. Recently, new tools have become available for efficiently visualising 3D volumetric data [7–9], and some of those allow the user to overlay tracking data to cross-check the quality of the results or to visualise simple predefined features (such as speed or time). However, given the more general-purpose design of such software, these are not ideal solutions to efficiently and collaboratively explore and share the visualisations.

Tools like CATMAID [10] or Neuroglancer (https://github.com/google/neuroglancer) impressively demonstrated the benefit of in-browser 3D visualisations for collaborative curation and visualisation of neuroimaging data [11]. In contrast to the specialised focus of these tools on volumetric neuroimaging data (e.g. reconstructing and visualising neural morphologies from electron microscopy images), we aimed to build a general-purpose, lightweight, and interactive visualisation of generic trajectory data across all fields of biology that might be challenging to visualise in static images otherwise (from animal tracks or static brain tractography to cellular or molecular motion). Here, interactive, scriptable, and easily shareable visualisation [12] open up novel ways of communicating and discussing experimental results and findings [13]. The analysis of complex trajectory data and the creation and testing of hypotheses could then be done collaboratively. Importantly, since such bioinformatics tools would be right at the interface of computational and life sciences, they need to be accessible and usable for scientists with little or no background in programming. Ideally, the data should be bundled with a guided, interactive presentation in one concise packet that can be passed to a collaborator. To address these challenges, we have developed our tool *linus*, making it easier to explore 3D trajectory data from any device without a local installation of specialised software. *linus* creates interactive visualisation packets that can be explored in a web browser while keeping data presentation straightforward and shareable, both offline and online (Fig 1A). In previous work, we explored cell trajectories during zebrafish gastrulation extracted from large-scale fluorescence microscopy datasets [2]. In these experiments, *linus* allowed us to interactively explore the tracks of around 11.000 cells (starting number) as they moved across the zebrafish embryo throughout 16 hrs. More importantly, it enabled us to share and discuss visualisations with collaborators from different backgrounds and to create figures for the manuscript.

By sharing this tool with the community, we hope to facilitate novel applications of visualising trajectories across all of biology. We have written this manuscript for a broad audience and thus mainly concentrate on describing how to create, use, and share the visualisations in the *Results* section from a user perspective. The *Design and Implementation* section and the S1 Text describe the technical details for readers who need to deploy the tool on their data. Finally, we refer readers interested in contributing new functionality or adapting the existing code (maintainers) to the technical documentation at our repository https://gitlab.com/imb-dev/linus.

## Design and implementation

From a more technical perspective, our overall goal when developing *linus* was to create a versatile and lightweight visualisation tool that runs on a wide range of devices. To this end, we based the visualisation part on web technologies. Specifically, we used TypeScript, which

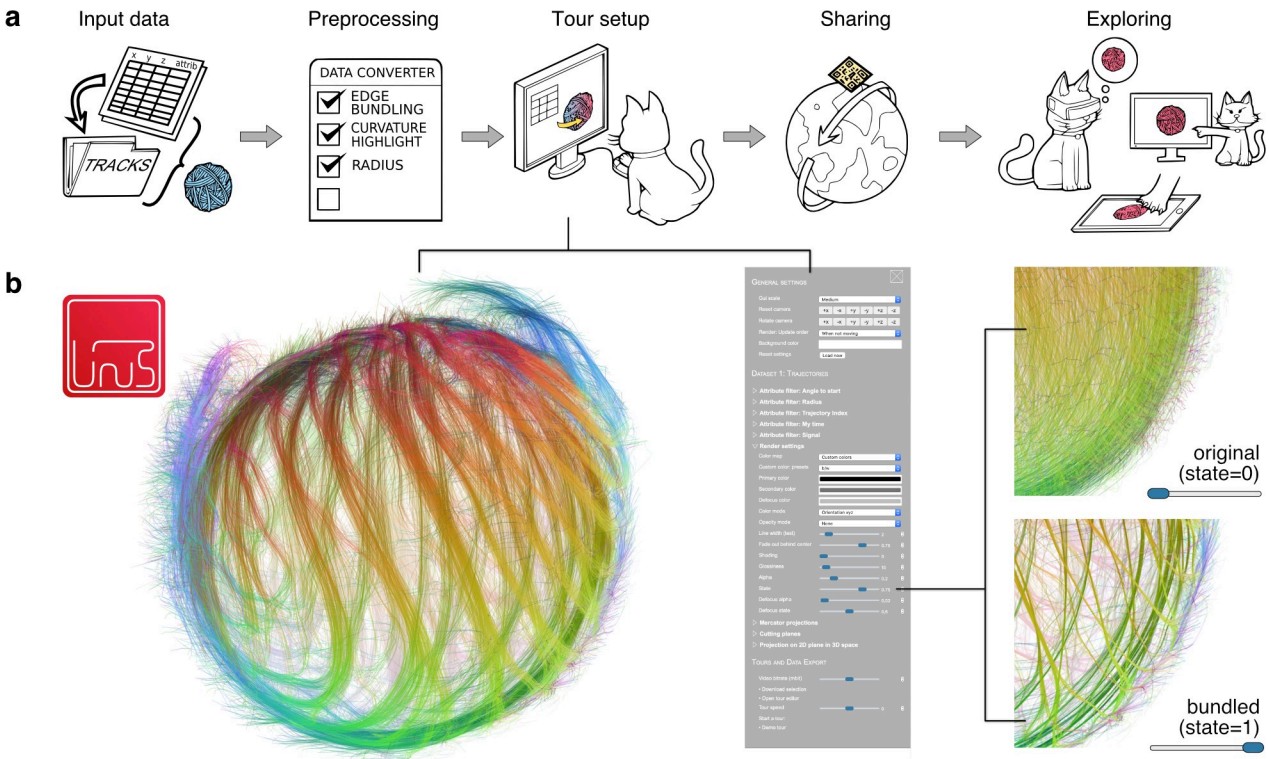

**Fig 1. Browser-based exploration and sharing of trajectory visualisations with *linus*.** (a) Control workflow of *linus*. (Input data) *linus* can import tracking data from a variety of formats. (Preprocessing) The Python-converter additionally enriches the imported data with additional features (providing e.g. an edge-bundled version of the data, visual context, or a coordinate system) and prepares the visualisation packet. (Tour setup) The user can open the visualisation in a web browser and create an interactive presentation of the data. (Sharing) These visualisations can be shared via a URL, or a QR code and (Exploring) readily presented and explored across various devices. (b) Overview of the graphical user interface (GUI). The data can be visualised and explored in the browser. Different aspects of the data can be interactively highlighted (zoomed example on the right shows the effect of changing the degree of trajectory bundling).

compiles to JavaScript and WebGL. Moreover, a core component of the visualisation process, the data preparation, requires local file access and fast computations, both of which are limited in JavaScript. For that reason, we also created a Python ($>$ v3.0) script that handles the computationally demanding parts of data processing and automatically generates the web-based visualisation packets.

The overall workflow for creating visualisations is summarised in Fig 1A. The importer script from *linus* can read trajectory data from a generic, plain CSV format (see S1 Text) or from a variety of established trajectory formats such as SVF [5], TGMM XML [14], or the community standard biotracks [15], which itself supports import from a wide variety of cell tracking tools such as CellProfiler [16] or TrackMate [17]. During the data conversion, *linus* can enrich the trajectory data with additional attributes or spatial context. For example, we can declutter dense trajectories by highlighting the major "highways" through edge-bundling (Fig 1B). *linus* can automatically add generic attributes that are useful in a range of applications, such as the local angle of the trajectories or a timestamp. The user can simply add custom numerical attributes for specific applications by providing these measurements as extra columns in CSV files (see S1 Text). The data attributes form the basis for advanced rendering effects. To highlight the spatial context, *linus* can generate axes automatically, or users can define custom axes. The result of the preprocessing is a ready-to-use visualisation packet that

can be opened in a web browser on any device with WebGL support. The packet is a folder containing HTML, JavaScript, and related files.

After configuring and creating the core visualisation with the Python toolkit, further adjustments are possible within the web browser. Opening the index.html file starts the visualisation and shows the trajectories with baseline render settings (semi-transparent, single-coloured rendering on a grey background). The browser renders an interactive visualisation of the trajectories and an interface for the user to update and adapt the presentation to their needs (e.g. colour scales, projections, clipping planes). The user interface itself is adapted to each dataset: The preprocessing script generates a separate property and the corresponding slider (filters and colour mapping) for each given data attribute in the user interface. If more than one state is available for the dataset (e.g. an edge bundled copy of the data or custom projections), the interface automatically offers the functionality to fade between the states. An in-depth discussion of the technical details can be found in the S1 Text and in our online documentation.

## Results

### Interactive visualisation with configurable filters allows in-depth data exploration for a variety of applications across sciences

After configuring and creating the visualisation packet with the Python toolkit, the user can carve out patterns from the original "hairball" of lines by setting general visualisation parameters like shading and colour maps (Fig 2A). For example, the user can directly encode the local movement direction (x, y, z) into RGB values by selecting the colour mode to *orientation XYZ*. To further enhance the visibility of movement directions, the visibility of parts of the trajectory can be gradually reduced by the *Opacity mode* option that maps, e.g. the time dimension to the opacity channel. To focus on particular parts of the dataset, the user filters the data for the various attributes such as specific time intervals or user-specified numerical properties such as marker expression in cell tracking (Fig 2B and 2C). Alternatively, the user can select spatial regions of interest (ROIs) either with cutting planes or with progressively refinable selections (Fig 2D). The visual attributes can then be separately defined for the selected in-focus areas and the (non-selected) context regions (Fig 2D) to create a more focused visualisation. Apart from the purpose of qualitative visualisation, the selected trajectories can also be downloaded as CSV files for subsequent quantitative analysis (see S1 Text).

One crucial problem with large-scale trajectory data is the sheer density of tracks that often leads to extreme visual clutter. To tackle this problem, one prominent feature of *linus* is the ability to blend between different data transformations seamlessly. We provide two main sorts of transformations out-of-the-box: The user can smoothly transition between original and bundled state to focus on major "highways" (Figs 2D and 1B), or between original (3D cartesian) view and different 2D projections (e.g. a Mercator map) to provide a global, less cluttered perspective on the trajectories (Fig 2E and 2F). If other application-specific transformations are needed, such as a spatial transformation or any form of trajectory clustering, the user can provide such an alternative state during preprocessing and then interactively blend between those states.

### Data and visualisations are easily shareable with collaborators via interactive visualisation packets

As a straightforward solution to share the results, the user directly exports the visualisations from the webview as static images and videos (e.g. such as S1 Video). But sharing the visualisation of the data can go a step beyond image or video data. The user can conveniently record all

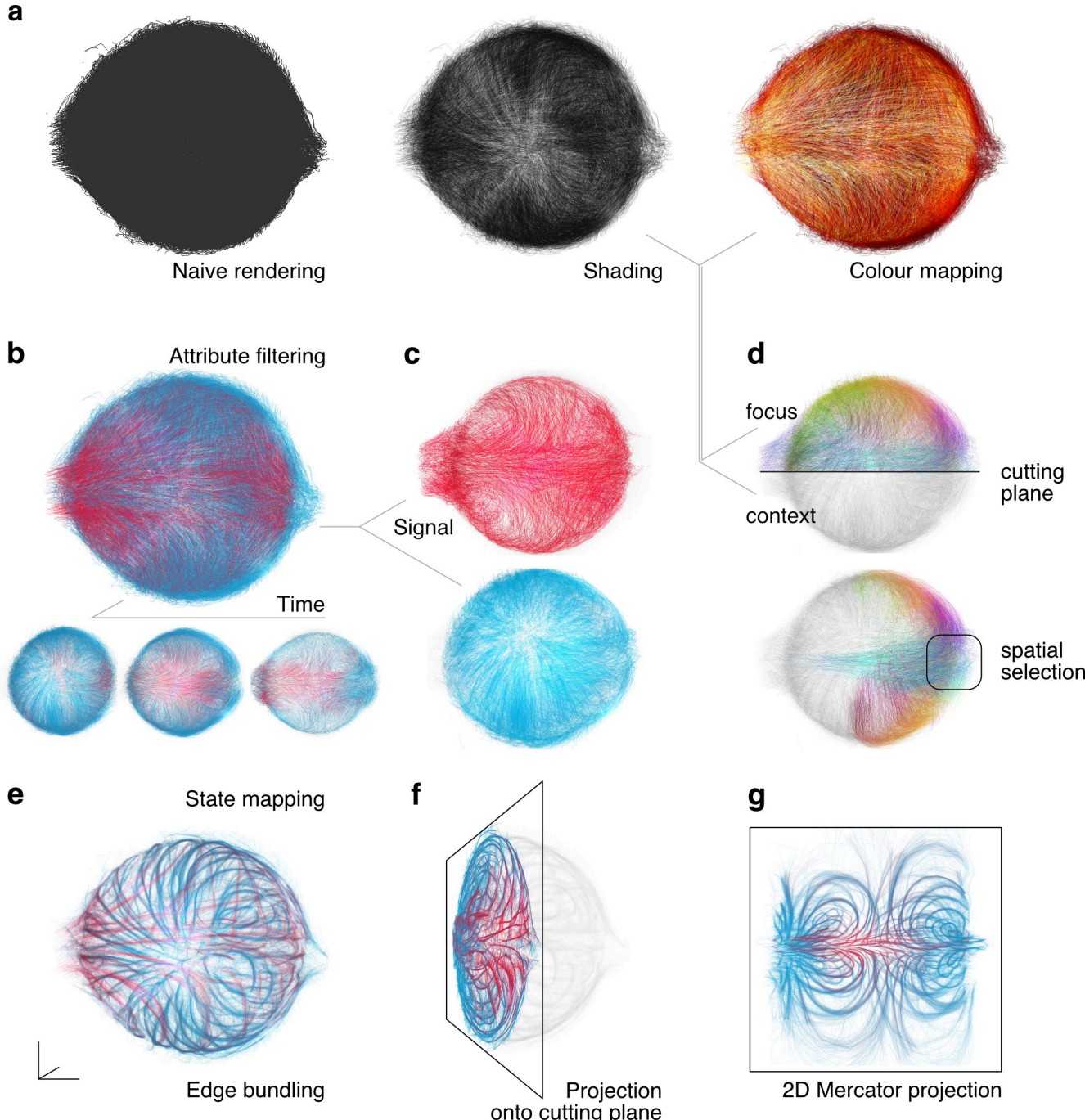

**Fig 2. Configurable filters allow deep data exploration.** The user can choose from a range of several visualisation methods directly in the browser interface to highlight aspects of interest in the data (zebrafish tracking results from [2] as an example). (a) The line data is visualised using a range of options for shading and colour mapping. (b-c) From the full dataset (top), the user can filter parts of the data concerning specific attributes, such as time intervals (bottom) or (c) a specific range of signals (marker expression in cells in this case). (d) The user can further create subselections of the tracks in space using cutting planes or refinable spatial selections. The visual attributes can be defined separately for the selected focus region and the non-selected context region. (e-g) The web interface can blend seamlessly between different states of the data. This feature can be used to map between (e) original tracks and their edge-bundled version, to visualise planar projections of the 3D data (f) locally on a definable (oblique) plane or (g) globally using a Mercator projection (with definable parameters).

these visualisation properties directly in the web interface of *linus* to create information-rich, interactive tours. The user adjusts these tours on a detailed level using a timeline-based editor (S1 Fig). An icon represents each action that can be moved along the time axis to develop a visual storyline. Smooth transitions and textual markers that can be precisely timed facilitate understanding and storytelling. To communicate and distribute new findings, these tours can easily be shared online or offline with the community (colleagues, readers of a manuscript, audience of a real or virtual presentation). The tours are typically copied into the source code of the visualisation packet. If they consist of a limited number of actions (see S1 Text for details), they can also be shared by a dynamically created URL or a QR Code. Fig 3 shows examples of visualisations that have been created with *linus* ranging from dynamic trajectories in 2D (Fig 3A) or on surfaces (Fig 3B) to static (Fig 3C) or dynamic 3D (Fig 3D) tracks across applications from ethology, neuroscience, and developmental biology. An interactive version of each example can be found online by simply scanning the respective QR codes in the figure.

We tested *linus* visualisation packets across various devices and found that performance is the most essential aspect of the user experience that varies between different devices. Desktop computers with mid-range graphics cards (e.g. the graphics processors that are built-in with current CPUs) can easily handle more than 10,000 trajectories at smooth framerates. Mid-range smartphones handle the same data with low framerates (ca. 10 fps), which is still usable but does not feel as smooth. For virtual reality applications, we also tested *linus* on the Oculus Go VR goggles. Here, a high frame rate is essential as the user experience would be quite discomforting otherwise, and we recommend reducing the number of trajectories further to about 1,000 in this use case. Due to the differences in performance and user experience, we recommend creating dedicated visualisation packets (or tours) for the intended type of output device.

## Availability and future directions

Example visualisations are available by scanning the QR codes in Fig 3 directly or by visiting https://imb-dev.gitlab.io/linus-manuscript/. The *linus* software, including source code and documentation, is freely available at our repository at https://gitlab.com/imb-dev/linus.

In the future, we would like to support further advanced preprocessing options such as trajectory clustering, more generic transforms or feature extraction. We also would like to extend the visualisation part of *linus*, so the user can interactively annotate the data. Here, we envision that the user can easily label subsets of trajectories and then use this information for downstream analysis (such as building a trajectory classifier). We also invite the community to contribute further ideas to develop *linus* or integrate its functionality as plugins to other tools. To contribute, please see the notes at https://gitlab.com/imb-dev/linus/-/blob/master/CONTRIBUTING.md.

Our experience with *linus* shows that sharing relatively complex data visualisations in this interactive way makes it much more efficient to collaboratively find patterns in data and to create and discuss figures or videos for presentations and manuscripts. More generally, interactive data sharing is helpful when collaborations, presentations, or teaching occur remotely. At the same time, during an in-person event such as a talk or poster session at a conference, the target audience can explore the data instantly on their computers, tablets, or smartphones. In any case, touch screens or even virtual reality goggles increase the immersion with more natural controls and true 3D-rendering, helping to grasp the trajectories' spatial relation. With these features, we are convinced that approaches like *linus* will improve considerably how we collectively explore, communicate, and teach the spatio-temporal patterns from information-rich, multi-dimensional data in biology.

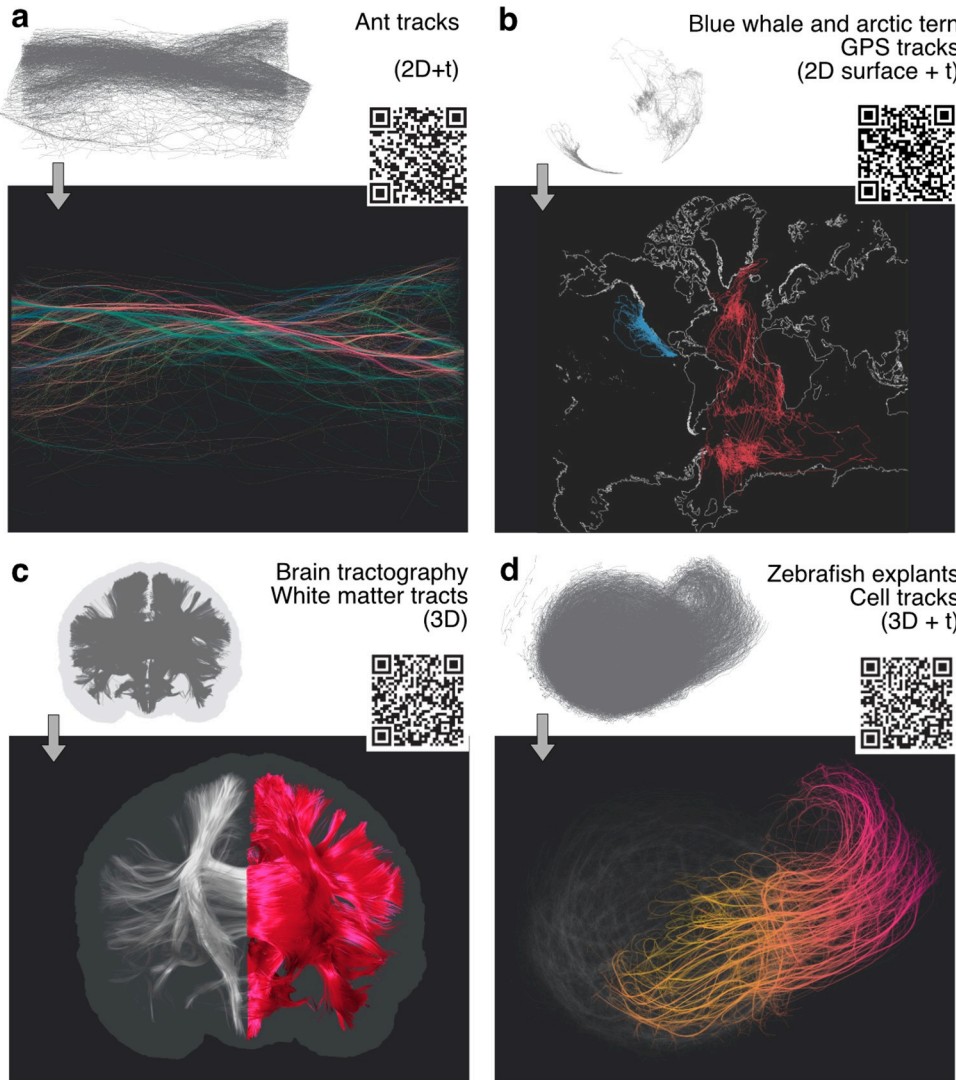

**Fig 3. Sharable interactive visualisation packets for a multitude of applications ranging across a variety of sciences.** The user can combine the visualisation methods, annotations, and camera motion paths in a scheduled tour that can be shared by a custom URL or QR code generated directly in the browser interface. Panels (a)-(d) demonstrate use cases for real-world datasets with different characteristics and dimensionality. (a) Ant trails (2D+t) from [18]. Bundling and colour-coding (spatial orientation by mapping (x,y,z) to (R,G,B) values) indicate the major trails running in opposing directions. (b) GPS Animal tracking data for two species (blue whales [19]—blue and arctic tern [20]—red) shown on a Mercator projection of the earth's surface. For a better orientation, the outline of the continents is included as axes into the visualisation that dynamically adapt to the projections and viewpoint changes (2D surface data + t). (c) Brain tractography data showing major white matter connectivity from diffusion MRI (3D). The spatial selection highlights the left hemisphere, while anatomical context is provided by the outline of the entire brain (from mesh data) and the defocused tracts of the right hemisphere. (d) Cell movements during the elongation process of zebrafish blastoderm explants (3D+t) [21]. Bundling, colour coding, and spatial selection highlight collective cell movements as the explant starts elongating, focusing on a subpopulation of cells driving this process. The colour code shows time from early (yellow) to late (red) for selected tracks.

## Supporting information

**S1 Fig. Tour editor.** The tour actions can be organised by drag and drop (reading order: from left to right, top to bottom). Every action can be scheduled with a time delay with respect to the end of the previous action. Some actions use transitions (e.g. camera motions or the

adjustment of numeric values) whose duration can be configured as well. Eventually, a URL or a QR code can be created.
(TIFF)

**S2 Fig. Overview of data structure.** The coordinate list holds the x/y/z values for each supporting point of the trajectories. For each such point, an arbitrary number (only limited by the graphics card's capabilities) of attributes can be stored. The attributes must be provided in the same order as the points. To create trajectories from the point set, an index list is provided as well. Each pair of indices describes one segment of a trajectory. The number of such segments is not restricted, as any point (and its respective attributes) can be used multiple times.
(TIFF)

**S3 Fig. Overview of settings.** An overview of the different visualisation settings available to the user from the GUI (two screenshots merged). For explanations regarding different settings, see text or documentation at https://gitlab.com/imb-dev/linus.
(TIFF)

**S1 Video. A video showing a demo visualization created and recorded with *linus*.** The example shows 3D + t trajectories of cells during elongation of zebrafish blastoderm explants (see also Fig 3D).
(MP4)

**S1 Text. Supplementary text discussing more technical details of the design and implementation of *linus*.**
(DOCX)

## Acknowledgments

The authors are grateful to Gopi Shah and Konstantin Thierbach for sharing data and contributing helpful feedback.

## Author Contributions

**Conceptualization:** Mario Hlawitschka, Jan Huisken, Nico Scherf.

**Data curation:** Johannes Waschke, Kerim Anlas, Vikas Trivedi, Jan Huisken, Nico Scherf.

**Investigation:** Johannes Waschke, Nico Scherf.

**Methodology:** Johannes Waschke, Mario Hlawitschka, Nico Scherf.

**Resources:** Kerim Anlas, Vikas Trivedi, Jan Huisken.

**Software:** Johannes Waschke, Nico Scherf.

**Supervision:** Mario Hlawitschka, Vikas Trivedi, Ingo Roeder, Jan Huisken, Nico Scherf.

**Visualization:** Johannes Waschke, Nico Scherf.

**Writing – original draft:** Johannes Waschke, Kerim Anlas, Vikas Trivedi, Jan Huisken, Nico Scherf.

**Writing – review & editing:** Johannes Waschke, Mario Hlawitschka, Kerim Anlas, Vikas Trivedi, Ingo Roeder, Jan Huisken, Nico Scherf.

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
