## [Decision Letter · Decision Letter 0]

2 May 2021

Dear Dr Scherf,

Thank you very much for submitting your manuscript "linus: Conveniently explore, share, and present large-scale biological trajectory data from a web browser." for consideration at PLOS Computational Biology.

As with all papers reviewed by the journal, your manuscript was reviewed by members of the editorial board and by several independent reviewers. In light of the reviews (below this email), we would like to invite the resubmission of a significantly-revised version that takes into account the reviewers' comments.

We cannot make any decision about publication until we have seen the revised manuscript and your response to the reviewers' comments. Your revised manuscript is also likely to be sent to reviewers for further evaluation.

Sincerely,

Manja Marz

Software Editor

PLOS Computational Biology

Manja Marz

Software Editor

PLOS Computational Biology

Reviewer's Responses to Questions

**Comments to the Authors:**

Reviewer #1: The authors present a new web toolkit for visualizing 3D data. The need for tools that can perform these functionalities is great in neuroscience right now. And they have built something that they seem to be using profitably. However, I have a few serious concerns, having been involved in writing software like this many times before.

1. Writing software like this is difficult, maintaining it is much more difficult. In the last 10 years there have probably been at least 10 published neuroscience visualization toolboxes developed, several of them for local clients, but also several of them browser based. Every single one failed unless it was developed by an institution (such as Google or Janelia). All the ones developed by labs died. The reason is that browser infrastructure and graduate students both change so far. Any toolkit will fail to work in a browser relatively quickly (say, ~2 years) without active developments, because the browsers change. And graduate students graduate, meaning that somebody else must take over the maintenance of the code. But graduate students like developing their own stuff, and also, they should develop their own stuff, because they need to if they want to graduate (they usually do). So, while I have no doubt that this tool is useful within the lab, I have serious evidence-based concerns about its utility more broadly. If (and this is a big if) the authors had a serious commitment (including financial commitment) for at least 5 years with a software engineer (rather than graduate student) who was committed to staying for 5 years, then I'd recommend the authors discuss the level of commitment extensively in the article, as tools without maintenance are not particularly useful.

2. The tool actually does a few different things. First, it is a toolkit to enable visualization of fibers *at all*. Second, it is a set of features/functions/add-ons that enhance the visualization capabilities. It is actually the former that so worries me in my above comments. The latter could be generally useful, and even maintained long-term relatively easily, if they were plug-ins to already widely adopted software. For example, Neuroglancer (which was not cited) is currently used by hundreds of people. While my team is not actively developing NeuroGlancer, we were developing our own related tool until Neuroglancer came out, and then we switched. Fortunately, it seems there is lots of overlap between very Neuroglancer capabilities, and your functionality and code. For example, both are written primarily in TypeScript, and both have a Python API. While Neuroglancer was originally developed for electron microscopy data, people certainly visualize tracts using it as well. CATMAID can also render tracts in 3D. There are several other widely use web-browser based tools with overlapping functionality that are not mentioned at all. So, it is not clear the extent to which the functionality proposed here is novel/unique, nor it is clear how easy it would be to have features/plug-ins to other existing established tools, rather than building yet another neuroscience visualization engine. Here are some links to Neuroglancer that show some relevant functionality:

https://layer23.microns-explorer.org/

https://www.youtube.com/watch?v=zqjJkFmLauE

Much more in depth justification of a new tool rather than a fork/feature/add-on to NeuroGlancer, CATMAID, or other existing browser based web-visualizers would be required.

3. Is it actually useful? There is no evaluation of its practical utility. Who currently uses it? Why? Is there empirical data one can point to? I found it difficult to control and navigate, and I did not understand the controls.

4. There are 3 potential target audiences:

A. People who simply use the website to visualize some data. They need to understand user features, but nothing else.

B. Moderators: these people need to be able to deploy the code as well, ingest data, etc., and therefore understand more of the guts of the code.

C. Maintainers: these people need to be able to modify the code itself to add functionality, etc.

It is not clear to me exactly which of these audiences this manuscript is written for, it feels like it fluctuates between these three groups. Please make it clear to whom you are writing for any given paragraph, so it is clear.

Reviewer #2: J. Waschke et al. present a visualisation tool to investigate two- and three-dimensional trajectory data. The easy-to-use application allows to examine trajectories from all angles, modify the visual representation and select data subdomains for more in-depth examination in a web-based interface. The authors nicely place their tool in the context of a growing amount of tracking data and interdisciplinary research that would benefit from quick sharing of results such as visualizations.

Linus is a very useful application and many scientists will profit from this easy-to-use tool. The user interface is well designed, with beautiful visualizations and an intuitive workflow. The paper is well written and easy to understand. The authors clearly state which problems Linus is intended to solve. The documentation of the github repository contains all the relevant details and guides the user well. We found linus easy to apply from the git-repository and it worked as described on Mac on our own data. We particularly like the idea of shareable QR codes to switch between devices seamlessly. Finally, besides sharing visualizations with collaborators, the tool comes in handy to produce beautiful figures for manuscripts and talks.

We strongly support the publication of the tool and suggest addressing the issues below.

* Issues:

** Data handling:

- Data import should be more flexible. From a TrackMate .csv file we got the error “ValueError: could not convert string to float: '88.862;113.354;40'’. We solved this by using a comma (“,”) instead of a (“;”) as a separator and converting the numbers. As this might be a recurring error for other users the authors should make the data loader more flexible and robust and thereby easier to use.

- We were not able to visualize multiple tracks contained in one .csv file as generated from TrackMate (e.g. with columns: Label, ID, TRACK_ID, QUALITY ,POSITION_X, POSITION_Y, POSITION_Z, POSITION_T ,FRAME). Can you direct the user with a few how tos / a video tutorial for the major use cases / data formats?

Please add examples of files that can be imported and use cases to the github, which the user can follow. This would be very helpful.

** The tool:

- Selecting and highlighting single tracks would be extremely useful.

- Highlighting track directions between timepoints (in polar, spherical or cartesian coordinates) would be great, so one can easily explore movement patterns.

- Pls comment: How is the software going to be sustained?

* Minor issues:

** Wording:

- We suggest to use ‘often’ only once in the first sentences of abstract and into

- Not clear what is meant by ‘camera’ when first used on page 8(? please add page numbers)

- British vs American english: e.g. color vs colour, visualization vs visualisation

- Typo in Fig. 2 caption: “(d) The user can further create subselections of the tracks in space using cutting planes or refinable spatial selections.” Should be selections at the end of the sentence?

- Type in Fig 3: ‘tracts’

- Fig 3a: either ‘ant migration’ or ‘animal tracks’ we suggest.

** Structure:

- Last sentences of the intro (“We began…”) feel a bit misplaced. Use the zebrafish example as the motivation?

The authors mixed up subfigures in Figure 3: (e) and (f) should be exchanged with each other and changed to (c) and (d).

- “Exemplary visualizations are available by scanning the QR codes in Fig.1” - It should be Fig. 3 here

paragraph ‘We tested linus…’ is misplaced in ‘Availability and future directions’ we believe. Better separate into two sub chapters

** Presentation:

- Fig 1b looks great, also for a cover, but maybe change to an example that reminds the reader less of toupee? ;-)

- Caption Fig. 1: Pls describe the steps in words, one by one

move ‘Furthermore, we cannot provide…” sentence to a “Limitations of the approach” paragraph?

- Fig 2: Not clear to me what the thin gray lines mean. Am I supposed to read from left to right? c to b?

**Have the authors made all data and (if applicable) computational code underlying the findings in their manuscript fully available?**

Reviewer #1: Yes

Reviewer #2: Yes

PLOS authors have the option to publish the peer review history of their article (what does this mean?). If published, this will include your full peer review and any attached files.

Reviewer #1: No

Reviewer #2: **Yes: **Carsten Marr with Valerio Lupperger and Dominik Waibel
---

## [Decision Letter · Decision Letter 1]

30 Sep 2021

Dear Dr Scherf,

We are pleased to inform you that your manuscript 'linus: Conveniently explore, share, and present large-scale biological trajectory data in a web browser.' has been provisionally accepted for publication in PLOS Computational Biology.

Best regards,

Manja Marz

Software Editor

PLOS Computational Biology

Jason A. Papin

Editor-in-Chief

PLOS Computational Biology

Reviewer's Responses to Questions

**Comments to the Authors:**

Reviewer #2: The authors addressed all raised issues and improved their work considerably. In particular, they updated the instructions (e.g. the step-by-step tutorial), improved the tool (e.g. opacity that can be coupled with time), fixed the trackmate .csv file import, and addressed tool maintenance.

We support publication of the manuscript.

Minor suggestions:

- Single track highlighting works nicely until settings (e.g. Mercator projections -> Rotation x) are changed. Only after closing the slider (e.g. of Mercator projections) it works again.

- Selecting a track bundle (from Render->State=1) would be probably helpful and nice to share with collaborators.

**Have the authors made all data and (if applicable) computational code underlying the findings in their manuscript fully available?**

Reviewer #2: Yes

PLOS authors have the option to publish the peer review history of their article (what does this mean?). If published, this will include your full peer review and any attached files.

Reviewer #2: **Yes: **Carsten Marr

---

## [Editor Report · Acceptance letter]

27 Oct 2021

PCOMPBIOL-D-21-00164R1 

linus: Conveniently explore, share, and present large-scale biological trajectory data in a web browser.

Dear Dr Scherf,

I am pleased to inform you that your manuscript has been formally accepted for publication in PLOS Computational Biology. Your manuscript is now with our production department and you will be notified of the publication date in due course.

With kind regards,

Livia Horvath
